# Sickle Cell Hemoglobin Genotypes Affect Malaria Parasite Growth and Correlate with Exosomal miR-451a and let-7i-5p Levels

**DOI:** 10.3390/ijms24087546

**Published:** 2023-04-19

**Authors:** Keri Oxendine Harp, Alaijah Bashi, Felix Botchway, Daniel Addo-Gyan, Mark Tetteh-Tsifoanya, Amanda Lamptey, Georgina Djameh, Shareen A. Iqbal, Cecilia Lekpor, Saswati Banerjee, Michael D. Wilson, Yvonne Dei-Adomakoh, Andrew A. Adjei, Jonathan K. Stiles, Adel Driss

**Affiliations:** 1Department of Physiology, Morehouse School of Medicine, Atlanta, GA 30310, USA; kerioxendine99@yahoo.com (K.O.H.); abashi@msm.edu (A.B.); shareen.iqbal@gmail.com (S.A.I.); sbanerjee@msm.edu (S.B.); 2Department of Pathology, Korle-Bu Teaching Hospital, University of Ghana Medical School, Accra P.O. Box 4236, Ghana; felixbotchway@yahoo.com (F.B.); cella20gh@yahoo.com (C.L.); andrewanthonyadjei@yahoo.com (A.A.A.); 3Department of Parasitology, Noguchi Memorial Institute for Medical Research, University of Ghana, Accra P.O. Box LG581, Ghana; danextreme4jc@gmail.com (D.A.-G.); mark.tetteh@ymail.com (M.T.-T.); amandalamptey24@gmail.com (A.L.); ginabel200@gmail.com (G.D.); mwilson@noguchi.ug.edu.gh (M.D.W.); 4Department of Haematology, Korle-Bu Teaching Hospital, Accra P.O. Box 77, Ghana; deiadom@yahoo.com; 5Department of Microbiology, Biochemistry and Immunology, Morehouse School of Medicine, Atlanta, GA 30310, USA; jstiles@msm.edu

**Keywords:** malaria, sickle cell disease, parasitemia, red blood cells, microRNA, exosomes, extracellular microvesicles

## Abstract

Malaria affects a significant portion of the global population, with 247 million cases in 2021, primarily in Africa. However, certain hemoglobinopathies, such as sickle cell trait (SCT), have been linked to lower mortality rates in malaria patients. Hemoglobin (Hb) mutations, including HbS and HbC, can cause sickle cell disease (SCD) when both alleles are inherited (HbSS and HbSC). In SCT, one allele is inherited and paired with a normal allele (HbAS, HbAC). The high prevalence of these alleles in Africa may be attributed to their protective effect against malaria. Biomarkers are crucial for SCD and malaria diagnosis and prognosis. Studies indicate that miRNAs, specifically miR-451a and let-7i-5p, are differentially expressed in HbSS and HbAS compared to controls. Our research examined the levels of exosomal miR-451a and let-7i-5p in red blood cells (RBCs) and infected red blood cells (iRBCs) from multiple sickle Hb genotypes and their impact on parasite growth. We assessed exosomal miR-451a and let-7i-5p levels in vitro in RBC and iRBC supernatants. Exosomal miRNAs exhibited distinct expression patterns in iRBCs from individuals with different sickle Hb genotypes. Additionally, we discovered a correlation between let-7i-5p levels and trophozoite count. Exosomal miR-451a and let-7i-5p could modulate SCD and malaria severity and serve as potential biomarkers for malaria vaccines and therapies.

## 1. Introduction

In 2021, malaria affected 247 million people, with the vast majority (95%) of cases occurring in Africa [1]. Although it is not clear why individuals develop either uncomplicated or severe *Plasmodium falciparum* (*Pf*) infections [2,3], genetic factors may play a role in the severity of the disease in individuals with certain hemoglobinopathies [2]. For instance, individuals with sickle cell trait (SCT) and thalassemia trait experience a lower malaria mortality rate [2,4,5,6]. SCT and sickle cell disease (SCD) result from genetic mutations in the beta chains of the hemoglobin (Hb) gene, leading to morphological changes in red blood cells (RBCs) in hypoxic conditions [7]. When both alleles are mutated, HbS or HbC variants cause sickle cell anemia (SCA) (HbSS or HbSC). In SCT (HbAS or HbAC), one variant allele (HbS or HbC) is inherited from one parent, and a normal allele (HbA) from the other [8,9]. Individuals with HbCC genotypes are also known to have some degree of protection against malaria [4]. However, those with SCA have a higher mortality rate than those with a normal Hb genotype (HbAA) or SCT when infected with malaria [5]. Furthermore, SCA patients have a lower malaria prevalence and parasitemia [4]. Previous studies have also shown significant differences in inflammation levels, white blood cell (WBC) counts, and RBC counts based on different sickle Hb genotypes with and without malaria [10,11]. Nonetheless, underlying the molecular mechanisms of protection observed in SCT remain to be explored.

MicroRNAs (miRNAs) are small, non-coding RNAs that endogenously up- or down-regulate targeted gene expression post-transcriptionally [12]. MiRNAs are produced by various cells, and a single miRNA can have multiple targets [13]. MiRNA profiles differ between resting and disease states, thus making them potential diagnostic and prognostic biomarkers [13]. MiRNAs could be used as biomarkers in body fluids for several pathologies, including cancer and infectious diseases [14,15,16,17]. In plasma, miRNAs are preserved in exosomes (microvesicles (MVs) of 30 to 100 nm in size) [18]. Exosomes can be transferred from one cell to another [19]. Exosomal miRNAs have been used to develop new diagnostic and therapeutic tools in cancer and chronic or infectious diseases [18,20,21,22]. Since exosomal miRNAs can easily be accessed in plasma and other body fluids, and are stable for long-term storage, they make excellent biomarkers [14]. However, there is limited information about the role of miRNAs in SCD and malaria.

*Plasmodium* parasites enter the human body through a female *anopheles* mosquito bite and release sporozoites that travel to the liver. In the liver stage, schizonts are formed and break open, releasing merozoites into the bloodstream, which causes the symptoms of malaria. During the blood stage, immature trophozoites develop into mature trophozoites or gametocytes, which have five phases and can be transmitted to the female mosquito during a blood meal [23].

Interestingly, *Plasmodium* lacks the enzymatic machinery necessary for miRNA production [14]. Nevertheless, studies have demonstrated that double-stranded RNA can downregulate gene expression, suggesting a non-canonical RNA interference mechanism that targets mRNA molecules to inhibit gene expression or translation [14]. Therefore, it is imperative to further investigate the interaction and function of miRNAs in *Plasmodium* parasites. Previous studies have shown that miR-451a and let-7i-5p levels were elevated in HbSS and HbAS erythrocytes and negatively regulated parasite growth in vitro [9,24]. Additionally, a separate study found a correlation between miR-451 and parasite densities in *Plasmodium vivax* [25]. In our study, we hypothesized that host sickle Hb status and exosomal miR-451a and let-7i-5p levels in serum mediate malaria pathogenesis by regulating parasite growth and survival.

## 2. Results

### 2.1. Variations in Overall Parasite Count Observed among Different Sickle Hb Genotypes

Blood samples were collected from 24 participants without *P. falciparum* infection, and there were four participants per genotype (HbAA, HbAS, HbAC, HbSC, HbSS, HbCC) (Appendix A). Parasites were grown in RBCs from participants with different sickle Hb genotypes and had similar patterns of overall growth during the 16 days of measurement (Figure 1). Area under the curve (AUC) measurements demonstrated that HbAA had the greatest parasite growth count and HbCC had the lowest. We also found that the mean of the two groups differed significantly via Tukey’s multiple comparison test (*p* = 0.03). We completed post hoc multiple comparison tests to determine group differences while controlling for Type I error inflation [26]. 

Data from daily parasite counts were analyzed on days 3, 8, 9, and 16 (Appendix A). On day three, parasite counts were highest for HbAC and HbSC, followed by HbAS, HbAA, HbCC, and HbSS groups. Interestingly, the highest parasite counts were recorded in HbAA and HbCC on day eight. Nevertheless, on day eight, despite a similar pattern to day three, there were no significant differences between sickle Hb genotypes. 

It is interesting to note that the parasite count for HbSS was the lowest on days three, eight, and nine. HbCC had the lowest parasite count on day 16 (Appendix A), while HbSS had the lowest parasite count on all the other days. 

### 2.2. Variation in Parasite Stage Distribution among Different Sickle Hb Genotypes

To understand parasite phase distribution, we synchronized *Pf* cultures so that only the ring stage was present at the beginning of the experiment. Figure 2 shows the distribution of distinct morphological phases of the parasite count, including rings, trophozoites, schizonts, and gametocytes, and indicates a slight difference in phase distribution between sickle Hb genotypes. Throughout the study, rings were consistently observed (Figure 2A), while trophozoites were primarily observed during the first eight days (Figure 2C), and schizonts were mostly observed after day six (Figure 2E). The average sum of each parasite phase during the 16-day period was estimated using AUC analysis. We found that trophozoite numbers were similar between all sickle Hb genotypes, except for HbCC, for which a lower number of trophozoites (Figure 2D). Interestingly, HbSS had more schizonts than the other sickle Hb genotypes, while HbCC had the least count of schizonts (Figure 2F).

Comparison of parasite numbers by phase based on genotype for days 3, 8, 9, and 16 (Appendix A) showed that regardless of the sickle Hb genotype, day 3 was primarily composed of trophozoites. Furthermore, the phases of gametocytes were observed each day (Figure 3). According to previous research [23], gametocytes can be classified into five distinct morphological phases (phases 1-5) in RBCs. Although only a small number of gametocytes were observed early in the culture, the majority were found after day six. During the 16 day culture period, the overall presence of each gametocyte phase was estimated using AUCs. The HbAA genotype had the highest number of gametocytes in all phases, while HbCC had the fewest phase-one gametocytes (Figure 3B). Phase-one gametocytes peaked on day ten for all sickle Hb genotypes. HbSS and HbCC had the lowest number of phase-two gametocytes, which also peaked on day ten (Figure 3D). HbCC had the fewest phase-three gametocytes, followed by HbSS (Figure 3F). Interestingly, HbAA, HbAS, and HbSC had higher counts of phase-four gametocytes than HbAC, HbSS, and HbCC (Figure 3H). The peak counts for phases three and four gametocytes varied depending on the sickle Hb genotypes. Phase-five gametocytes are the ones that are picked up by mosquitoes from their hosts. HbCC had the lowest number of phase-five gametocytes, while HbAA and HbSC had the most (Figure 3J). Phase-five gametocytes peaked on day sixteen, and only phase-four showed significant (*p* < 0.05) differences between sickle Hb genotypes (Appendix A). In HbAA, phase-four gametocytes were the most abundant (mean = 14.5%) (Appendix A), and their counts were significantly higher than those of HbAC (mean = 3.875%, *p* = 0.01), HbSS (mean = 5.5, *p* = 0.04), and HbCC (mean 2.75%, *p* = 0.004) genotypes. Additionally, we examined whether parasite growth was correlated with sickle Hb genotype over time (Table 1). Interestingly, all sickle Hb genotypes were significantly (*p* < 0.006) correlated with trophozoite counts. Moreover, there was a significant (*p* < 0.05) correlation between all sickle Hb genotypes and phases four and five gametocytes, indicating that sickle Hb genotypes influence the development of parasites, specifically for trophozoites and gametocytes at phases four and five.

### 2.3. Alterations in Exosomal Levels of let-7i-5p and miR-451a Observed in P. falciparum-Infected Individuals with Different Sickle Hb Genotypes

This study analyzed the exosomal levels of let-7i-5p and miR-451a in infected red blood cells (iRBCs) and non-infected RBCs at different time points (days 3, 8, 9, and 16) during a 16 day culture (as shown in Figure 1 and Figure 2). The AUC was calculated for each sickle Hb genotype with and without malaria to estimate the exosomal miRNA levels. Although no significant differences were found in exosomal miR-451a or let-7i-5p levels between sickle Hb genotypes with (+) and without (-) malaria, HbSS- had higher levels of exosomal let-7i-5p (as shown in Figure 4D) compared to HbSC- or HbAA- (as shown in Figure 3B). In addition, both HbAS- and HbAC- had lower exosomal let-7i-5p levels. Alternatively, *Plasmodium* infection of RBCs resulted in lower let-7i-5p levels in HbAS+, HbAC+, HbSS+, and HbSC+ sickle Hb genotypes, while HbAA+ had the lowest levels of let-7i-5p (as shown in Figure 4D). When comparing parasite count and exosomal let-7i-5p levels at different time points, a Pearson correlation test revealed a significant correlation on day eight (R^2^ = 0.29, *p* = 0.02, as shown in Figure 4F). Exosomal miR-451a levels were slightly higher in HbSS- compared to other sickle Hb genotypes without malaria (as shown in Figure 4I), but all sickle Hb genotypes showed a similar trend in overall exosomal miR-451a levels when RBCs were infected with malaria, as with let-7i-5p (as shown in Figure 4L).

The exosomal miRNA levels for iRBCs and RBCs with sickle Hb genotypes were evaluated every day (Figure 4). An ANOVA/Tukey’s multiple comparison test on day eight showed that exosomal miR-451a was significantly higher in HbSC- compared to HbSS- (*p* = 0.02). Furthermore, miRNA levels were compared between RBCs and iRBCs for each genotype on days 3, 8, 9, and 16 (Appendix A). Exosomal miR-451a levels were significantly (*p* = 0.03) higher for HbSS+ compared to HbSS-, and exosomal let-7i-5p levels at day eight were significantly (*p* = 0.04) higher for HbAC+ compared to HbAC- (Figure 5B, Appendix A).

Exosomal miR-451a and let-7i-5p were compared on days 3, 8, 9, and 16 to assess their correlation (Figure 5). We observed a significant correlation between exosomal miR-451a and let-7i-5p levels in RBCs on days 3 (R^2^ = 0.48, *p* = 0.0003), 8 (R^2^ = 0.61, *p* < 0.0001), 9 (R^2^ = 0.24, *p* = 0.02), and 16 (R^2^ = 0.45, *p* = 0.009). However, on day eight, we found that trophozoite counts were negatively correlated with exosomal let-7i-5p levels (R^2^ = 0.26, *p* = 0.02), indicating that higher trophozoite counts were associated with lower exosomal let-7i-5p levels (data not shown). Ultimately, while exosomal miR-451a and let-7i-5p levels were positively correlated in RBCs, this correlation was not observed when the RBCs were infected.

## 3. Discussion

To our knowledge, this is the first study to examine *Pf* parasite growth in relation to exosomal miR-451a and let-7i-5p levels in different sickle Hb genotypes. By examining exosomal miRNA levels in iRBCs in vitro instead of an in vivo model, we gain insight into what the miRNA levels are in RBCs regardless of the influence of other tissues, such as WBCs, vascular cells, and other blood components. This helps us understand how exosomes found in RBCs can influence *Pf* and vice versa. 

Other studies have shown that SCA patients have a higher mortality rate from malaria than HbAA or SCT patients [7]. Additionally, they had higher malaria rates during hospitalization than during outpatient clinics [4]. HbAS provides at least 90% protection against malaria mortality, and HbCC over 90% protection [27,28]. However, HbAC protected about 30% of patients and decreased parasite density [27,28,29], although with conflicting reports [28,30]. Lower parasite densities have been found in HbAS, HbAC, HbCC, HbSS, and HbSC patients [4,30,31]. When infected with malaria, HbSS and HbSC are at a disadvantage compared to HbAC, HbCC, and HbAS [28,30].

In this study, we observed 16 days of parasite growth in vitro in different sickle Hb genotypes. Based on AUC curves, HbAA had a significantly higher parasite count than HbSS and other sickle Hb genotypes over 16 days. HbCC had the lowest parasite count overall in both AUC and ANOVA tests. AUC data assessed parasite growth over time, while the one-way ANOVA multiple comparison test compared the means of the groups independently, revealing that the HbCC group had significantly lower overall parasite counts (Figure 1H,I). AUC curves also showed that HbAA had the most trophozoites, while HbSS had the most rings and schizonts (Figure 2). HbCC had the lowest overall parasite count, as it remained the lowest estimated ring, trophozoite, and schizont count. Therefore, our findings are consistent with previous reports, which suggest that there are biochemical and hematological differences between hemoglobin genotype and malaria disease. We showed that growth was highest in HbAA, followed by HbAS, HbAC, HbSC, and HbSS. We found that growth was lowest in HbCC, which shows that the HbC allele can be protective, as seen in other studies [32].

When we looked at specific days, HbAA did not consistently have the most parasites. We saw that HbAC and HbSC had the most parasites on day three and that overall parasite count peaked on day eight for HbCC, on day nine for HbAA, HbAS, HbAC, and HbSC, while HbSS had the lowest parasite count overall (Appendix A). Based on this, it appears that the rate of *Pf* growth is dependent on the sickle Hb genotype. Days 3, 8, 9, and 16 were selected to analyze exosomal miRNAs. The rationale behind choosing those days is that previous parasite count data showed that *Pf* growth peaked around days eight and nine before decreasing again. To ensure a comprehensive assessment of the entire experimental period, we chose one day at the beginning, two days during the peak, and one day at the end for analysis.

Next, each genotype was examined for gametocyte count. In humans, gametocytes appear 10 to 12 days after infection [23]. The first four phases of gametocyte development are considered immature and are sequestered primarily in the liver, RBCs, and spleen [33]. Gametocyte development is morphologically distinct in each phase [23]. Stage-five is the active stage in the mosquito. When a mosquito bites an infected person, it ingests stage-five gametocytes. These gametocytes then develop into ookinetes, which penetrate the mosquito’s midgut wall. The ookinetes then develop into oocysts, which are the infective stage of the parasite [23]. According to the AUC analysis of this study, HbAA produces the most phase-five gametocytes, suggesting further research on different sickle Hb genotypes is needed (Appendix A). As compared to other sickle Hb genotypes, HbAA consistently had the highest (AUC mean = 41.5) stage-five gametocyte formation over sixteen days, while HbCC had the lowest (AUC mean = 7.95) (Figure 3), making them the lowest reservoir. It is interesting to note that phase-four gametocytes were more abundant in HbAA than HbAC, HbSS, and HbCC (Figure 3D). The results show that HbAA would eventually have significantly more phase-five gametocytes than any other genotype if the culture was continued. This finding is important because the more phase-five gametocytes a person produces, the higher the likelihood of malaria transmission. Potential drugs can be developed to prevent parasite growth and reduce malaria spread by understanding why HbAA produces more phase-five gametocytes.

Previous in vitro studies have shown elevated levels of miR-451a and let-7i-5p in HbSS and HbAS erythrocytes. These studies have also shown that miR-451a and let-7i-5p inhibited parasite growth [9,24]. Despite this, we found some differences in exosomal miR-451a levels between sickle Hb genotypes with and without malaria. Let-7i-5p and miR-451a have also been associated with parasite density [16]. Hence, we examined whether their exosomal levels correlated with parasite counts [9]. We assessed the correlation between trophozoite counts and exosomal let-7i-5p levels on day eight and found a trend indicating that higher exosomal let-7i-5p levels are associated with fewer trophozoites. This suggests that elevated exosomal let-7i-5p levels may be involved in parasite replication and lead to reduced trophozoite formation. We also observed a trend of increased exosomal miR-451a levels with a reduction in parasite count, consistent with previous studies on endogenous miRNA. These findings suggest that miR-451a may play a role in *Pf* replication through modification of the molecular machinery found in the parasite. We found that exosomal miRNA levels differed between sickle Hb genotypes for iRBCs and RBCs (Figure 4) and that there was a slight trend in exosomal miRNA levels between sickle Hb genotypes in RBCs and iRBCs. Thus, exosomal miRNAs may promote parasite development in RBCs during malaria pathogenesis. 

Additionally, we examined the correlation between sickle Hb genotypes and parasite counts over the entire study. Despite finding no correlation between sickle Hb genotypes and parasite counts, certain parasite phases did show significant associations (Table 1). A significant relationship was shown between trophozoites and sickle Hb genotypes, suggesting that sickle Hb genotypes inhibit parasite replication. Gametocytes of phases four and five also correlated with sickle Hb genotypes. The relationship between sickle Hb genotypes and gametocyte production suggested that *Pf* produced gametocytes differently, which is dependent on Hb status. To reduce malaria transmission, further research into this topic is needed to understand why some people produce more gametocytes than others.

We also investigated whether these miRNAs were correlated, as we found in the population studies [16]. Uninfected RBCs were correlated on all days examined, with the same trend. As one miRNA level increased, so did the other, which is consistent with our previous findings [16]. In hemoglobinopathies, cytokine and miRNA levels can help understand malaria pathogenesis. Furthermore, we previously showed that different sickle Hb genotypes with or without malaria parasites have different levels of CXCL10, CCL3, IL-8, TNF-α, and IL-6 in the blood [10,11]. In this study, there are no other factors involved, such as WBCs releasing cytokines/chemokines, drug intake, or lifestyle. A closed in vitro environment affects parasite growth primarily through RBC shape and content. This study is also noteworthy because it requires collecting blood samples from patients with different sickle Hb genotypes, including rare forms such as HbC haplotypes.

Nevertheless, there are some limitations to be considered. We used four volunteers per Hb group, and more replicates would have yielded statistically significant results between parasite counts and exosomal miRNA levels. Samples from participants with sickle Hb genotypes are difficult to obtain. Second, it was a single-center study. More study replications and larger cohorts are needed.

The literature suggests that there are significant hematological and biochemical differences between sickle cell genotypes, which may result in varying oxygen delivery indices [34,35]. This is supported by previous research indicating that oxygen plays a role in malaria resistance [36]. Moreover, studies examining the impact of the beta-thalassemia trait on redox equilibrium and cytoskeleton structure, depending on the severity of the mutation, suggest that similar differences may exist between sickle cell Hb genotypes [37]. Given that oxidative stress and cytoskeleton features have been linked to malaria resistance, it is possible that these genotype-related differences may also play a role in sickle cells’ protective effects against malaria [38,39].

In conclusion, we have shown that exosomal miR-451a and let-7i-5p might contribute to parasite growth and multiplication. Targeted therapeutics toward exosomal miR-451a and let-7i-5p might regulate malaria growth and replication in RBCs. As a result, therapeutic development might reduce the malaria burden and decrease the disease severity or mortality among persons with sickle Hb genotypes that are more susceptible to rapid and high parasite replication.

## 4. Materials and Methods

### 4.1. Study Population

The ethical approval for this study was obtained from the respective institutions’ ethics review boards, which included Morehouse School of Medicine (MSM, Atlanta GA, USA) Institutional Review Boards, the University of Ghana’s (UG) College of Health Sciences at the Korle-Bu Teaching Hospital (Accra, Ghana), and the Noguchi Memorial Institute for Medical Research (NMIMR, Accra, Ghana). All participants provided informed written consent, and they were recruited from the Greater Accra Region in Ghana during June 2017 and June 2019 as part of an ongoing SCD and malaria study funded by the National Institutes of Health (NIH) and conducted in collaboration with UG at MSM. Experiments were carried out at NMIMR, and each participant was assigned a numerical anonymized code.

Inclusion criteria were individuals without malaria, aged > 16 years, HIV negative, with no high fetal Hb (HbF), a detailed complete blood count (CBC), involving 6 groups of sickle Hb genotypes, HbAA, HbAS, HbAC, HbSS, HbSC, and HbCC and both sexes. Samples were sex and aged grouped ranging from 25 to 35 years old. Individuals with malaria or HIV-positive, high HbF, or partial CBC data were excluded. A total of 24 participants (4 per group) met all the inclusion criteria (Appendix A). 

### 4.2. Laboratory Assessment of Blood Specimens

Blood samples were collected using Vacutainer ACD tubes (BD Bioscience, San Jose, CA). Sickle cell status was determined using a cellulose acetate membrane electrophoresis method with HbAA and HbSS as controls [40,41]. Participants completed a medical questionnaire, and hematological characteristics of blood samples were determined by CBC counts. Malaria status was determined using rapid diagnostic test (RDT) kits that detect both *Pf*-specific protein HRP2 and Pan (Pan LDH), which can detect multiple *Plasmodium* species. The absence of parasites in blood samples was confirmed using thin smear microscopy [41]. HIV status was determined using RDT (First Response HIV-1-2 Kits).

### 4.3. Culturing Pf 3D7

For malaria in vitro studies, *Pf* clone 3D7 is widely used [9,23,42]. Erythrocytes from participants with different sickle Hb genotypes were used for parasite cultivation. To culture the *Pf* parasites, RBCs were isolated from whole blood, washed to remove residual plasma and exosomes, and cultured following the WHO-FIND-CDC Malaria Methods Manual 2018, SOP 4.1 Preparation of Reagents and Media for Culture of Malaria Parasites [43]. AlbuMAX (Cat# 11021029, ThermoFisher, Waltham, MA, USA) was used as a substitute for human serum in culture media, as it does not contain exosomes, which can interfere with experiments. Moreover, AlbuMAX has been used extensively for culturing *Pf* [44]. General cultures were maintained at a hematocrit of 10%, while experimental assays were conducted at a hematocrit of 2%. Parasites were synchronized using Sorbitol (Cat# BP439500, ThermoFisher, Waltham, MA, USA), and RBCs were directly inoculated with the initial parasitemia of 0.5% for all experiments. The culture flasks were incubated at 37 °C and gassed with 5% CO_2_, 5% O_2_, and 90% N_2_.

### 4.4. Exosomal miRNA Isolation and Quantification

Exosomes were extracted from the supernatant of RBCs using the Total Exosome Isolation kit (from cell culture media) (Cat# 4478359, Invitrogen, Waltham, MA, USA) in accordance with the manufacturer’s guidelines [45]. To begin with, the cell culture supernatant was subjected to centrifugation at 2000× *g* for 30 min to eliminate debris. Subsequently, 0.5 volume of Total Exosome Isolation (from cell culture media) was added to the sample, and the mixture was vortexed and left overnight at 2–8 °C. The next day, the sample was centrifuged at 10,000× *g* for one hour at 2–8 °C. The supernatant was then removed, and the exosomes were resuspended in 1XPBS.

The first concentrate exosome pellet was suspended in 200 µL of 1XPBS, followed by RNA isolation from exosomes using the Total Exosome RNA/Protein Isolation kit (Cat# 4478545, Invitrogen, Waltham, MA, USA) in accordance with the manufacturer’s instructions for sections “1-Isolate RNA-Organic extraction, 2-Isolate RNA- Purify the RNA, 3-Bind the RNA, 4-Wash the RNA, and 5-Elute the RNA”. To elute RNA, 50 µL of Elution Solution was used, and the concentration and purity of RNA were determined using the Thermo NanoDrop 1000 Spectrophotometer (ThermoFisher). The samples were then stored at −80 °C until further analysis.

The RNA was reverse transcribed using the miRCURY LNT RT kit (Cat #: 339340) and miRCURY LNA SYBR Green PCR Kit (Cat #: 33936) by Qiagen (Hilden, Germany). RT-qPCR was carried out using Roche LightCycler 480 (Roche Applied Science, Penzberg, Germany) and CFX Real-Time PCR Detection Systems (Bio-Rad, Hercules, CA, USA). Primers for miR-451a, let-7i-5p, and U6 snRNA (which served as the internal control) were obtained from Qiagen.

### 4.5. Statistical Analysis

The relative fold change in miRNA expression was evaluated using the ΔCT method and normalized to U6 snRNA expression. The Livak method was used to calculate ΔCt values in MS Excel (Microsoft, Redmond, WA, USA), with U6 snRNA as an endogenous control [46]. The data were normalized by log transformation, and statistical analysis was conducted using an unpaired *t*-test or one-way ANOVA with Tukey’s multiple comparison test.

All statistical analyses were performed using GraphPad Prism version 9.4.0 for Windows (GraphPad Software, La Jolla, California, USA) unless otherwise specified. The normality of the data was assessed using the D’Agostino–Pearson normality test. If the data did not pass the normality test and could not be transformed, nonparametric tests were used. A significance level of *p* < 0.05 was considered for all analyses. For data that passed the normality test, either one-way ANOVA and Tukey’s multiple comparison test for comparisons between more than two groups or an unpaired *t*-test for comparisons between two groups were used to test for significance. For data that did not pass the normality test (e.g., WBC counts), a Kruskal–Wallis test and a Dunn’s multiple comparison test or a Mann–Whitney test were used. Additional analyses included chi-squared, linear regression, Pearson correlation, receiver operating characteristic (ROC) curves, and the calculation of the area under the curve (AUC).

## Figures and Tables

**Figure 1 ijms-24-07546-f001:**
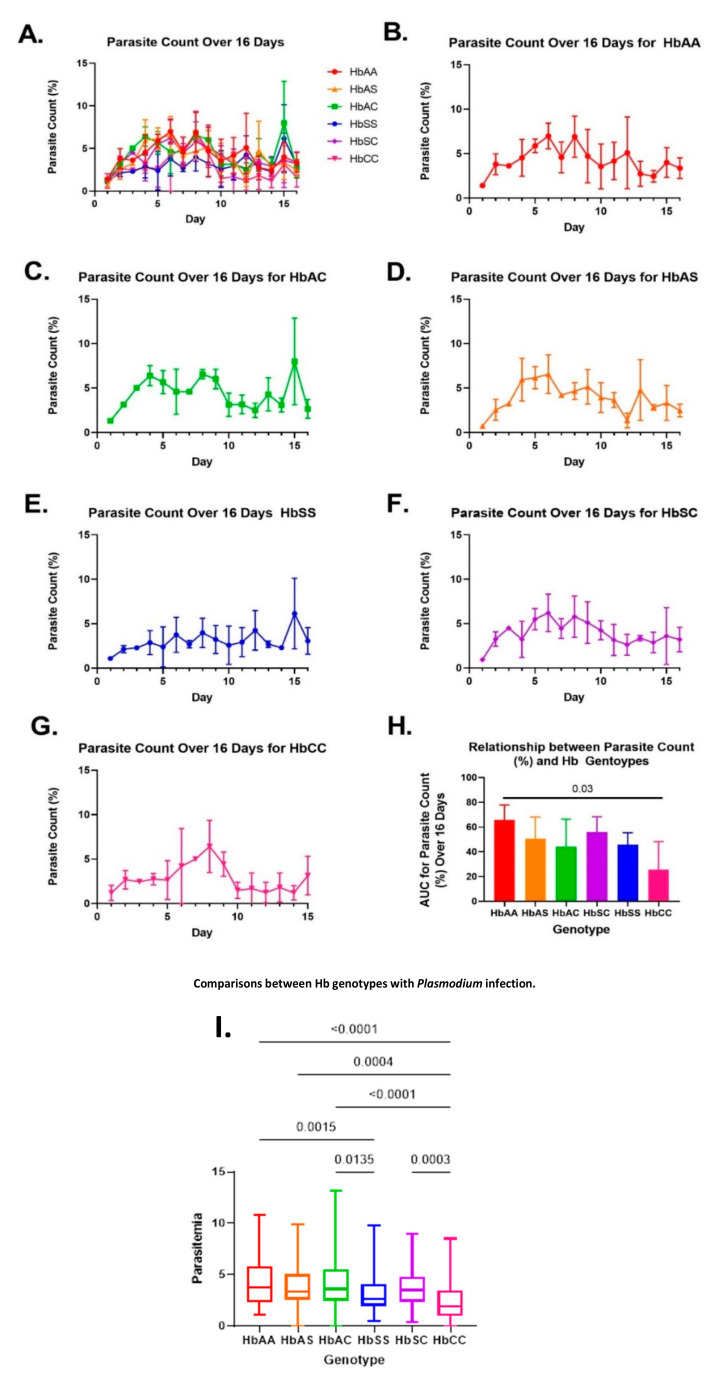
Parasite count percentage over 16 days culture of (**A**) all sickle Hb genotypes; (**B**) HbAA (red); (**C**) HbAC (green); (**D**) HbAS (orange); (**E**) HbSS (blue); (**F**) HbSC (purple); (**G**) HbCC (pink); and (**H**) area under the curve (AUC) for each genotype. AUC was used to estimate the overall parasite counts over 16 days. HbAA had significantly (*p* = 0.03) more parasites compared to the HbCC group. (**I**) A one-way ANOVA and Tukey’s multiple comparison test on all the genotypes. The median parasitemia was significantly lower in HbSS (*p* = 0.0015) and HbCC (*p* < 0.0001) groups compared to HbAA and in both groups compared to HbAC (*p* = 0.0135 and *p* < 0.0001, respectively). Additionally, the HbCC genotype had a significantly lower median parasitemia compared to HbAS and HbSC, with *p* = 0.0004 and *p* = 0.0003, respectively.

**Figure 2 ijms-24-07546-f002:**
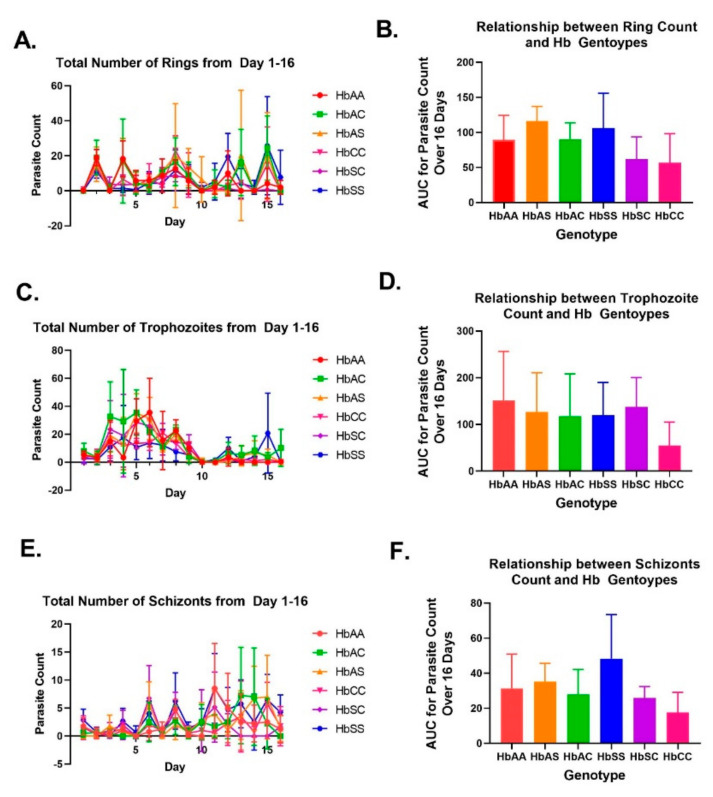
Ring, trophozoite, and schizont count for each sickle Hb genotype over 16 days. Red is for HbAA, orange for HbAS, green for HbAC, blue for HbSS, purple for HbSC, and pink for HbCC. Area under the curve (AUC) was used to estimate parasite counts over 16 days. A one-way ANOVA and a Tukey’s multiple comparison test were used. (**A**) Ring counts for each sickle Hb genotype. (**B**) AUC of overall ring count for each genotype. (**C**) Trophozoite counts for each sickle Hb genotype. (**D**) AUC for overall trophozoite count for each sickle Hb genotype. (**E**) Schizont counts for each sickle Hb genotype. (**F**) AUC for overall schizont count for each genotype.

**Figure 3 ijms-24-07546-f003:**
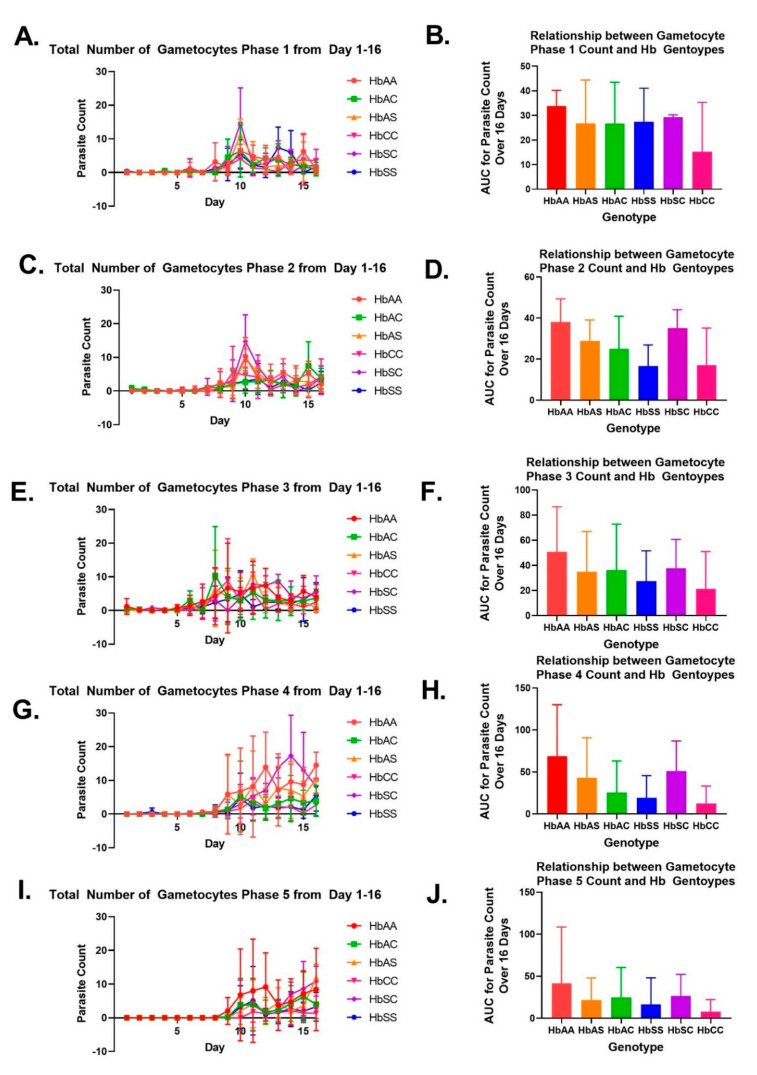
Gametocyte phase 1–5 counts for each sickle Hb genotype over 16 days. Red is for HbAA, orange for HbAS, green for HbAC, blue for HbSS, purple for HbSC, and pink for HbCC. Area under the curve (AUC) was used to estimate parasite counts over 16 days. A one-way ANOVA and a Tukey’s multiple comparison test were used. (**A**,**B**) illustrate phase-1 gametocyte counts for each genotype. (**C**,**D**) illustrate phase-2 gametocyte counts. (**E**,**F**) illustrate phase-3 gametocyte counts. (**G**,**H**) illustrate phase-4 gametocyte counts. (**I**,**J**) illustrate phase-5 gametocyte counts.

**Figure 4 ijms-24-07546-f004:**
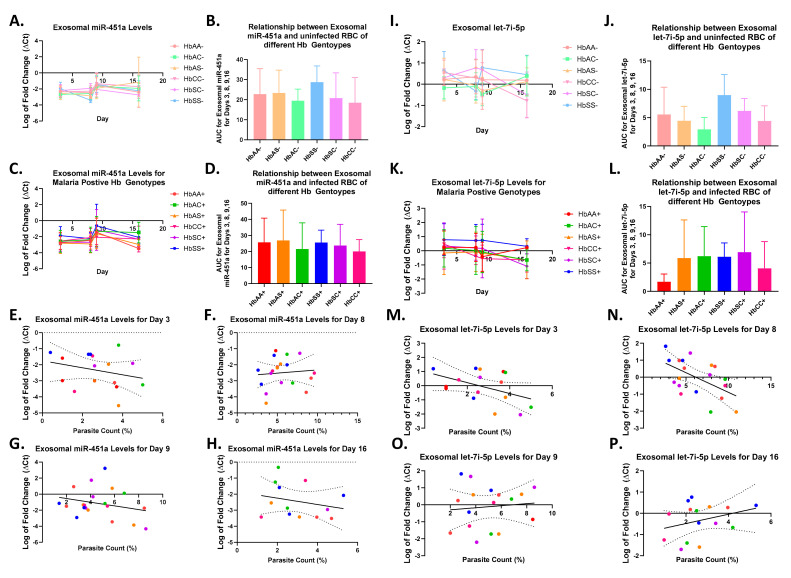
Exosomal miR-451a and let-7i-5p levels in *Plasmodium*-infected RBC and non-infected RBC of different sickle Hb genotypes. Area under the curve (AUC) was used to estimate exosomal miRNA levels over 16 days. A Pearson correlation was used for all correlations, and a linear regression was run. (**A**) shows the exosomal miR-451a levels of malaria-negative RBC with different hemoglobin (Hb) genotypes over 16 days. (**B**) presents the AUC for exosomal miR-451a levels over 16 days for each genotype without malaria. (**C**) displays the exosomal miR-451a levels of malaria-positive RBC with different sickle Hb genotypes over 16 days. (**D**) presents the AUC for exosomal miR-451a levels over 16 days for each sickle Hb genotype’s infected RBCs. (**E**–**H**) show the correlation between exosomal miR-451a levels and parasite count percentage on days 3, 8, 9, and 16, respectively. A significant positive correlation was observed on days 3 and 8 (R^2^ = 0.29 and *p* = 0.02). (**F**) includes the equation of the regression line (Y = −0.2316*X + 1.422). (**I**,**J**) display the exosomal let-7i-5p levels of malaria-negative RBC with different sickle Hb genotypes over 16 days and the AUC for exosomal let-7i-5p levels over 16 days for each genotype without malaria, respectively. (**K**) shows the exosomal let-7i-5p levels of malaria-positive RBC with different sickle Hb genotypes over 16 days. (**L**) presents the AUC for exosomal let-7i-5p levels over 16 days for each sickle Hb genotype’s infected RBCs. (**M**–**P**) show the correlation between exosomal let-7i-5p levels and parasite count percentage on days 3, 8, 9, and 16, respectively. The trend changes from a positive correlation on days 3 and 8 to a negative correlation on day 9, where let-7i-5p decreases with an increase in parasite count. (**O**) includes the equation of the regression line (Y = −0.2302*X − 0.08344).

**Figure 5 ijms-24-07546-f005:**
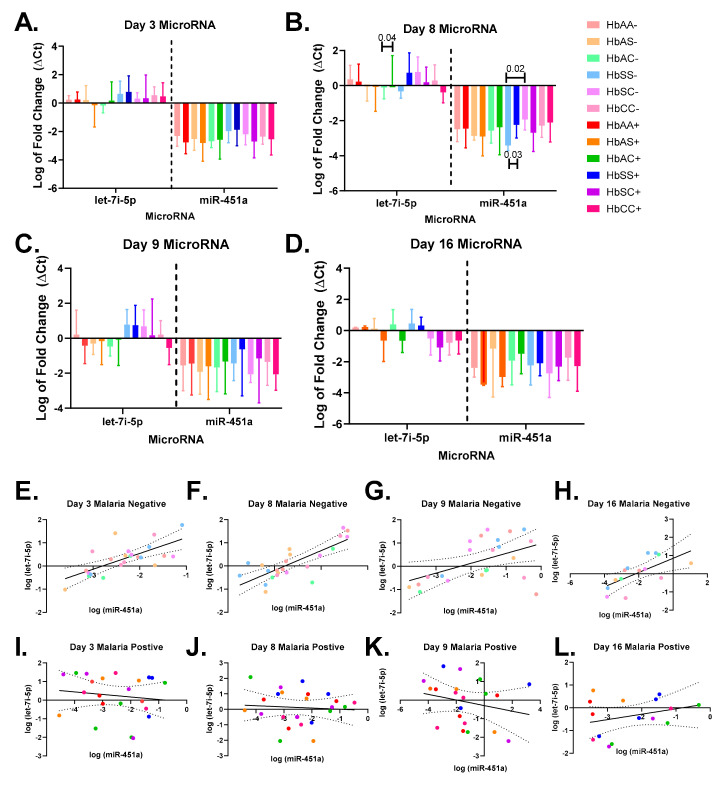
Exosomal miR-451 and let-7i-5p correlation and comparison between sickle Hb genotypes for days 3, 8, 9, and 16. Statistical analyses were performed using ANOVA and Tukey’s multiple comparison or Pearson correlation tests as appropriate, with significance determined for malaria-positive or negative sickle Hb genotypes unless stated otherwise. (**A**) displays miRNA gene expression on day 3, where no significant differences were found between malaria-positive or negative sickle Hb genotypes. In (**B**), miR-451a expression was significantly lower in HbSS- compared to HbSC- on day 8 (*p* = 0.02), while miR-451a was significantly higher in HbSS+ compared to HbSS (*p* = 0.03) and let-7i-5p was significantly higher in HbAC+ compared to HbAC- (*p* = 0.04) using an unpaired *t*-test. (**C**,**D**) demonstrate no significant differences in miRNA expression levels on days 9 and 16, respectively, for malaria-positive or negative sickle Hb genotypes. In (**E**–**H**), there were significant correlations found between exosomal miR-451a and let-7i-5p levels for malaria-negative sickle Hb genotypes on days 3, 8, 9, and 16, with respective equations of Y = 0.6803*X + 1.901, Y = 0.8521*X + 2.365, Y = 0.4235*X + 0.9760, and Y = 0.3904*X + 0.8477. The correlations had R^2^ values of 0.48, 0.61, 0.24, and 0.45 and were all significant with *p*-values of 0.0003, <0.0001, 0.02, and 0.009, respectively. (**I**–**L**) display no significant correlations between exosomal miR-451a and let-7i-5p levels for malaria-positive sickle Hb genotypes on days 3, 8, 9, or 16, respectively, with respective equations of Y = −0.1365*X – 0.1045, Y = −0.07589*X – 0.07005, Y = −0.1509*X – 0.2939, and Y = 0.2378*X + 0.1944. The *p*-values for all correlations were greater than 0.05. Additionally, the figures show the log-fold change in miRNA expression levels for days 3, 8, 9, and 16.

**Table 1 ijms-24-07546-t001:** Linear regressions for each genotype for overall parasite count and for each phase over 16 days.

Parasite Phase	Sickle Hb Genotype	R^2^, *p*-Value	Equation
Overall	HbAA	0.007, 0.51	Y = −0.04306*X + 4.645
	HbAC	0.0004, 0.9	Y = 0.01109*X + 4.156
	HbAS	0.07, 0.7	Y = −0.1243*X + 5.144
	HbSC	4.053 × 10^−5^, 0.96	Y = −0.002809*X + 3.956
	HbSS	0.08, 0.3	Y = 0.1234*X + 1.997
	HbCC	0.04, 0.22	Y = −0.08116*X + 3.263
Rings	HbAA	**0.08**, **0.02**	Y = −0.4903*X + 9.943
	HbAC	0.0001, 0.9	Y = 0.02736*X + 7.332
	HbAS	0.0003, 0.9	Y = −0.06652*X + 9.395
	HbSC	**0.19**, **0.0007**	Y = −0.5460*X + 8.440
	HbSS	0.06, 0.06	Y = 0.6375*X + 1.517
	HbCC	0.003, 0.72	Y = −0.1128*X + 6.341
Trophozoites	HbAA	**0.12**, **0.005**	Y = −1.089*X + 18.59
	HbAC	**0.14**, **0.009**	Y = −1.148*X + 21.47
	HbAS	**0.19**, **0.001**	Y = −1.274*X + 21.20
	HbSC	**0.16**, **0.0002**	Y = −1.150*X + 18.90
	HbSS	0.005, 0.57	Y = −0.1838*X + 8.988
	HbCC	**0.31**, **0.0001**	Y = −0.8259*X + 13.23
Schizonts	HbAA	0.04, 0.09	Y = 0.1672*X + 0.6515
	HbAC	0.06, 0.1	Y = 0.2371*X − 0.005741
	HbAS	0.13, 0.01	Y = 0.3098*X − 0.2333
	HbSC	0.0004, 0.88	Y = −0.01609*X + 2.068
	HbSS	**0.13**, **0.005**	Y = 0.3023*X + 0.7634
	HbCC	0.07, 0.08	Y = 0.1743*X + 0.09902
Phase-One Gametocytes	HbAA	**0.2**, **0.0002**	Y = 0.3219*X − 0.5794
	HbAC	0.05, 0.12	Y = 0.1844*X + 0.4737
	HbAS	0.06, 0.08	Y = 0.2160*X − 0.01239
	HbSC	0.06, 0.06	Y = 0.2687*X − 0.0385
	HbSS	**0.12**, **0.006**	Y = 0.2808*X − 0.5905
	HbCC	0.14, 0.01	Y = 0.2417*X − 0.7511
Phase-Two Gametocytes	HbAA	**0.19**, **0.0004**	Y = 0.3639*X − 0.6327
	HbAC	**0.17**, **0.004**	Y = 0.3254*X − 0.9284
	HbAS	**0.09**, **0.03**	Y = 0.2320*X + 0.003216
	HbSC	**0.09**, **<0.0001**	Y = 0.2067*X − 0.2117
	HbSS	0.1, 0.01	Y = 0.1711*X − 0.3202
	HbCC	0.08, 0.07	Y = 0.2067*X − 0.2117
Phase-Three Gametocytes	HbAA	**0.1**, **0.008**	Y = 0.3840*X + 0.08443
	HbAC	0.03, 0.24	Y = 0.2023*X + 0.9809
	HbAS	0.03, 0.2	Y = 0.1946*X + 0.7020
	HbSC	**0.28**, **<0.0001**	Y = 0.4977*X − 1.247
	HbSS	**0.14**, **0.004**	Y = 0.2872*X − 0.5419
	HbCC	0.03, 0.31	Y = 0.1289*X + 0.7532
Phase-Four Gametocytes	HbAA	**0.31**, **<0.0001**	Y = 0.9743*X − 3.516
	HbAC	**0.1**, **0.03**	Y = 0.3262*X − 0.9221
	HbAS	**0.24**, **0.0002**	Y = 0.6697*X − 2.554
	HbSC	**0.45**, **<0.0001**	Y = 0.9901*X − 4.328
	HbSS	**0.13**, **0.005**	Y = 0.2512*X − 0.7567
	HbCC	0.09, 0.05	Y = 0.1706*X − 0.3764
Phase-Five Gametocytes	HbAA	**0.16**, **0.001**	Y = 0.6156*X − 2.219
	HbAC	**0.14**, **0.009**	Y = 0.3842*X − 1.539
	HbAS	**0.3**, **<0.0001**	Y = 0.4824*X − 2.425
	HbSC	**0.34**, **<0.0001**	Y = 0.5963*X − 2.738
	HbSS	0.09, 0.02	Y = 0.2421*X − 0.8741
	HbCC	0.11, 0.03	Y = 0.1417*X − 0.5574

## Data Availability

The original contributions presented in this study are included in the article. Further inquiries can be directed to the corresponding author.

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
