# Peer review of "Sickle Cell Hemoglobin Genotypes Affect Malaria Parasite Growth and Correlate with Exosomal miR-451a and let-7i-5p Levels"

_ijms, 2023, doi:10.3390/ijms24087546_

Round 1

Reviewer 1 Report

As attached

Author Response

REVIEWER 1 COMMENTS:

Review comments

This is novel work that studied the correlation between exosomal MiRNA and malaria parasite growth in sickle cell haemoglobin genotypes.  The methods used were quite extensive towards answering the research questions or hypothesis. Nevertheless, to make the work better, here are few comments.

Abstract

Line 16- use the latest WHO report of 2022.

We have corrected the values from the WHO report 2022, now line 16-17.

Introduction

Line 34- There is recent Malaria report for 2021, check https://www.who.int/teams/global-malaria-programme/reports/world-malaria-report-2022

We have corrected the values from the new report and replaced the reference [1] accordingly in lines 36-37.

Line 61- diagnostic not “diagnostics”

Changed. Thank you! Now line 61.

Results

Line 96-106- The way the result in the figure is presented is not clear to the reader. It will be better to clearly define each Figure in the panel from A-I.  I suggest the title of the figure as follows:

Figure 1: Parasite count percentage overall of A. All Hb genotypes; B. HbAA; C. HbAC; D. HbAS; E. HbSS; F. HbSC; G. HbCC; H. Area under the curve (AUC) for each genotype and I. A one-way ANOVA and Tukey’s multiple comparison test on all the genotypes.

You can clearly put the explanations after this

We have fixed the Figure 1 legend accordingly. The new legend is (lines 100-108):

Figure 1. Parasite count percentage over 16 days culture of A. All Hb genotypes, ; B. HbAA (Red); C. HbAC (green); D. HbAS (orange); E. HbSS (blue); F. HbSC (purple); G. HbCC (pink); H. Area under the curve (AUC) for each genotype. AUC was used to estimate the overall parasite counts over 16-days. HbAA had significantly (p=0.03) more parasites compared to the HbCC group.  I. A one-way ANOVA and Tukey’s multiple comparison test on all the genotypes. The median parasitemia was significantly lower in HbSS (p=0.0015) and HbCC (p<0.0001) groups compared to HbAA, and in both groups compared to HbAC (p=0.0135 and p<0.0001, respectively). Additionally, the HbCC genotype had a significantly lower median parasitemia compared to HbAS and HbSC with p=0.0004 and p=0.0003, respectively.

What is the difference between figure H and I in terms of the data used?

Good point that needs some explanations to put the data in context. We have added these sentences to the discussion section in lines 277-282.

"Based on AUC curves, HbAA had a significantly higher parasite count than HbSS and other sickle Hb genotypes over 16 days. HbCC had the lowest parasite count overall in both AUC and ANOVA tests. AUC data assessed parasite growth over time, while the one-way ANOVA multiple comparison test compared the means of the groups independently, revealing that the HbCC group had significantly lower overall parasite counts (Figure 1H and 1I)."

Why did you choose to count the parasitemia on day 3, 8 and 9? Ideally, the parasite has a cycle every 48 hours. Would it not have been better if the parasite counts were done every cycle for the 16 days?

We have chosen those days (3, 8, 9 and 16)  for convenience. The total parasite count seemed to peak around days 8 and 9 and then goes down. So we have chosen a day in the beginning of the experiment, 2 days during the peak and one in the end so we can cover the overall experiment period. For this, we have added these sentences in the discussion Lines 294-298:

“Days 3, 8, 9, and 16 were selected to analyze exosomal miRNAs. The rationale behind choosing those days is that previous parasite count data showed that Pf growth peaked around days 8 and 9 before decreasing again. To ensure a comprehensive assessment of the entire experimental period, we chose one day at the beginning, two days during the peak, and one day at the end for analysis.”

Line 111- Were the parasites used for inoculating the RBCs synchronized? If yes, at what stage were they synchronized and what was the parasitemia used? Stating these will guide the reader in understanding the variation in the different phases of the parasite growth. I will suggest you include this information here.

Parasites were synchronized in this study for inoculation. We added this sentence in the methods, lines 114-115: “To understand parasite phase distribution, we synchronized Pf culture so that only the ring stage was present.”

For effective comparison of the phase distribution, it would be better to obtain the percentage of each phase at every time point in the course of the experiment. For instance, at day 8, if the overall parasite count is say 10, how many of this are rings, trophozoite, schizonts and gametocytes.  This percentage distribution for each genotype can enable you compare across other genotypes.

Supplemental figures 1, 2, and 3 compare parasite counts by each stage for days 3, 8, 9 and 16 between Hb genotypes. Explanations are in results in lines 95-99, 110-112, 135-161, and discussed lines 307-313.

Line 153-154- You are already discussing the work by citing a reference. Ideally, no references are allowed in result section. This is where you present your findings.

The reference was removed and text adapted accordingly, now lines 152-154.

Line 174-175 – correct the sentence

Corrected. New sentence, lines 181-184: “Although no significant differences were found in exosomal miR-451a or let-7i-5p levels between sickle Hb genotypes with (+) and without (-) malaria, HbSS- had higher levels of exosomal let-7i-5p (as shown in Figure 4D) compared to HbSC- or HbAA- (as shown in Figure 3B).

Line 186-215- Some of the information in Figure 2 had already been presented earlier. It amount to unnecessary repetition.

Corrected.

Line 230-251- Do same as suggested for figure 2

Figure 2 (new 3) and 3 (new 4) legend was re-written and reduced in length accordingly.

Discussion

Like 266- lie or like?

The sentence was removed, and text was adapted accordingly.

Line 298-299- Even though the results are presented in supplementary section, percentages of the gametocyte formation should be stated here. Refer to my suggestion about presenting your results above.

We have added the percentages in the text in lines 152-155 with the following:

“In HbAA, phase 4 gametocytes were the most abundant (mean=14.5%) (Supplemental Figure 2D), and their counts were significantly higher than those of HbAC (mean=3.875%, p=0.01), HbSS (mean=5.5, p=0.04), and HbCC (mean 2.75%, p=0.004) geno-types.”

And in the discussion line 309-311:

“As compared to other sickle Hb genotypes, HbAA consistently had the highest (AUC mean = 41.5) stage 5 gametocyte formation over 16 days, while HbCC had the lowest (AUC mean = 7.95) (Figure 3), making them the lowest reservoir.”

Methods

Line 381-390- No mention was made of the parasitemia used for the experiment

We added a sentence stating the initial 0.5% parasitaemia used. Line 415-417. “Parasites were synchronized, using Sorbitol and RBCs were directly inoculated with the initial parasitemia of 0.5% for all experiments.”

General comments

Most of the results are presented in supplementary Figures which are not accessible. It makes it difficult for reviewers to make input.

The supplemental figures will be added to the figures as a same file to avoid this kind of issues.

P- value is written as p value, with the p in lower case and italics

Changed.

Sentence correction and language check is necessary

Done. The overall text have been extensively changed to avoid plagiarism from the thesis and for better English.

Reviewer 2 Report

1. The authors should address few grammatical errors such as:

Line 48, ‘has’ should be ‘have’

Recast line 174-175

In line 266, ‘lie’ should be ‘like’

The sentence at line 276-277 is not okay

‘see’ at line 313 should be ‘saw’

2. The authors should address the plagiarism concern (see attached)

Author Response

REVIEWER 2 COMMENTS:

  1. The authors should address few grammatical errors such as:

Line 48, ‘has’ should be ‘have’

Line 49-51 Fixed: “Previous studies have also shown significant differences in inflammation levels, white blood cell (WBC) counts, and RBC counts based on different Hb genotypes with and without malaria [10,11].”

Recast line 174-175

Line 181-184- Rewritten to say: “Although no significant differences were found in exosomal miR-451a or let-7i-5p levels between sickle Hb genotypes with (+) and without (-) malaria, HbSS- had higher levels of exosomal let-7i-5p (as shown in Figure 4D) compared to HbSC- or HbAA- (as shown in Figure 3B).”

In line 266, ‘lie’ should be ‘like’

Line 266- The sentence was removed and text was adapted accordingly.

The sentence at line 276-277 is not okay

Lines 307-309- fixed to say: “According to the AUC analysis of this study, HbAA produces the most phase 5 gametocytes, suggesting further research on different sickle Hb genotypes is needed (Supple-mental Figure 3).”

‘see’ at line 313 should be ‘saw’

The sentences was removed and the text was adapted accordingly.

  1. The authors should address the plagiarism concern (see attached)

We have re-written all parts that had major plagiarism concerns from Dr. Harp’s thesis.

Reviewer 3 Report

In their manuscript "Sickle cell hemoglobin genotypes affect malaria parasite growth and correlate with exosomal miR-451a and let-7i-5p levels" Keri Oxendine Harp et al., examine Plasmodium growth in distinct sickle cell genetic backgrounds and controls, as well as the occurrence of exosomal miRNA in each culture. The study is interesting and can help us understand the link between Hb mutations and malaria resistance, but it needs some improvements before its consideration for publication.

Abstract (minor comments):

1) I strongly believe that the first 5 lines of the Abstract concern general knowledge. I think that the SCD alleles and their link to malaria resistance can be written in fewer lines.

2) Please give a definition for iRBCs when first mentioned in the Abstract section

Introduction:

Minor comments

1) In the first paragraph of the Introduction section the authors state that distinct Hb genotypes have been linked to malaria resistance. I understand that their research is focused on SCT, but I think that thalassemia traits should also be mentioned in this section.

2) In line 54 instead of writing up or down regulate, please write "up- or down-regulate" to make it easier for the reader to follow the sentence.

3) In line 57 the authors mention the diagnostic value of miRNAs, I think they should also mention their usefulness as prognostic biomarkers.

Major comment:

4) I think information regarding the life cycle of Plasmodium sp. is missing from the Introduction section. Since the authors provide information regarding distinct phases of the parasite's life cycle (gametocyte, trophozoites etc.) it is important to mention these phases in the Introduction, especially since the life cycle of Plasmodium is not the easiest to remember. In the Results section there are some pieces of information regarding gametocytes, but I believe this part should be enriched with more information, with regards to the whole sexual reproduction cycle of the parasite, and then transferred to the Introduction section.

Methods:

Minor comment:

1) Were RBCs washed before their culture with parasites to make sure that there is no residual plasma (and therefore exosomes)?

Major comment:

2) The authors state both in the Methods and the Results section that they performed a multiple comparison test. I don't understand how this was possible since there are 6 groups with 4 subjects each, leading to not enough degrees of freedom to perform multiple comparisons. Could they be clearer regarding the groups they compared? Did they perform multiple comparisons only in groups that provided statistically significant results from ANOVA? 

Results:

Minor comments:

1) To facilitate the reader, I believe that the authors should break Supplementary Figure 1 into two Figures. Since Supplementary Figure 1 panels E-P are mentioned after Supplementary Figure 2, I would suggest that the authors move these panels to a novel Supplementary Figure (namely, Supplementary Figure 3; and change the current Supplementary Figure 3 to Supplementary Figure 4) so that the reader doesn't have to go back and forth when reading the article.

2) In line 137 the authors state "on day 16, mainly gametocytes were observed" and afterward, line 141 starts with "the phases of gametocytes were also determined". I think the authors should first mention gametocytes when their results are shown so that the reader can follow the manuscript.

3) I think that Table 1 should be transferred near the section it is first mentioned.

Major comments:

4) Figure 2 is really hard to follow. Since I understand why it is structured in this way, I think the authors could add a line in the middle to separate the Results regarding the one miRNA from the other and maybe add a title above each side (e.g., the names of the miRNAs) to help the reader follow the Figure. 

5) In all Figures legends the authors should remove the sentences "there was (not) a significant difference/correlation" and rather add a sign of significance on the diagrams, since the legends are extremely extensive and really hard to follow.

6) A general comment: The authors spend a lot of time writing about Results that are not statistically significant. Please refrain from comparing values with no significance. For example, in lines 131-132 it is written "Still on day 8, there were no significant differences in schizont counts between the sickle Hb genotypes, but HbSS had the most and HbAS the least". Since there is no statistical significance, I don't find it proper to mention which genotype has the most and which the least. Or, at least, the authors are kindly asked to refrain from doing in excessively in the manuscript.

Discussion (major comment):

Many parts of the Discussion section are just a brief repetition of the Results. I understand that in a manuscript with so many comparisons, it is useful to repeat the main findings in the Discussion, but I believe that the authors need to add more pieces that interpret their findings. In my opinion, they should try to explain their findings based on the current bibliography, and not just state that they are in line with previously reported data. For example, it is known that there are hematological and biochemical differences between these genotypes (10.1371/journal.pone.0228399), as well as a distinct oxygen delivery index (10.3233/CH-2008-1142). Indeed, oxygen has been found to play a role in malaria resistance (10.1073/pnas.1804388115). Moreover, in a similar background, namely beta-thalassemia trait, the different genotypes result in different redox equilibrium and distinct cytoskeleton structure depending on the severity of the mutation (10.3389/fphys.2022.907444). Could be this also true for the different sickle cell genotypes? Especially since both oxidative stress and cytoskeleton features have been linked to malaria resistance (10.1038/ncomms13401; 10.1002/(SICI)1096-8644(199906)109:2<269::AID-AJPA11>3.0.CO;2-#)? I believe that the authors could add some bibliographically-based hypotheses to explain their Results, like the suggestions above. Of course, they are free to search and reference more relative articles.

Overall: There are many grammatical/syntax errors in the manuscript. The authors are kindly requested to proofread their Revised manuscript before its resubmission. 

Author Response

REVIEWER 3 COMMENTS:

Abstract (minor comments):

  • I strongly believe that the first 5 lines of the Abstract concern general knowledge. I think that the SCD alleles and their link to malaria resistance can be written in fewer lines.

It has been rephrased to make it clearer. Lines 16-20: “Malaria affects a significant portion of the global population, with 247 million cases in 2021, pri-marily in Africa. However, certain hemoglobinopathies, such as sickle cell trait (SCT), have been linked to lower mortality rates in malaria patients. Hemoglobin (Hb) mutations, including HbS and HbC, can cause sickle cell disease (SCD) when both alleles are inherited (HbSS and HbSC). In SCT, one allele is inherited and paired with a normal allele (HbAS, HbAC).”

2) Please give a definition for iRBCs when first mentioned in the Abstract section

Fixed. Lines 23-25: “Our research examined the levels of exosomal miR-451a and let-7i-5p in red blood cells (RBCs) and infected red blood cells (iRBCs) from multiple Hb genotypes and their impact on parasite growth.

Introduction:

Minor comments

  • In the first paragraph of the Introduction section the authors state that distinct Hb genotypes have been linked to malaria resistance. I understand that their research is focused on SCT, but I think that thalassemia traits should also be mentioned in this section.

We have added the Thalassemia trait to the text. Line 39-41: “For instance, individuals with sickle cell trait (SCT) and Thalassemia trait experience a lower malaria mortality rate [2,4-6].”

2) In line 54 instead of writing up or down regulate, please write "up- or down-regulate" to make it easier for the reader to follow the sentence.

Fixed. Lines 54-55: “MicroRNAs (miRNAs) are small, non-coding RNAs that endogenously up- or down- regulate targeted gene expression post-transcriptionally [12].”

  • In line 57 the authors mention the diagnostic value of miRNAs, I think they should also mention their usefulness as prognostic biomarkers.

Fixed. Lines 56-58: “MiRNA profiles differ between resting and disease states, thus making them potential diagnostic and prognostic biomarkers [13].

Major comment:

  • I think information regarding the life cycle of Plasmodium sp. is missing from the Introduction section. Since the authors provide information regarding distinct phases of the parasite's life cycle (gametocyte, trophozoites etc.) it is important to mention these phases in the Introduction, especially since the life cycle of Plasmodium is not the easiest to remember. In the Results section there are some pieces of information regarding gametocytes, but I believe this part should be enriched with more information, with regards to the whole sexual reproduction cycle of the parasite, and then transferred to the Introduction section.

Lines 66-71: added the following: “Plasmodium parasites enter the human body through a female anopheles mosquito bite and release sporozoites that travel to the liver. In the liver stage, schizonts are formed and break open, releasing merozoites into the bloodstream which causes the symptoms of malaria. During the blood stage, immature trophozoites develop into ma-ture trophozoites or gametocytes, which have five phases and can be transmitted to the female mosquito during a blood meal.”

Methods:

Minor comment:

1) Were RBCs washed before their culture with parasites to make sure that there is no residual plasma (and therefore exosomes)?

Yes, RBCs were washed 3X using RPMI to remove residual plasma and exosomes. A sentence was added in the 4.3 results paragraph. Lines 407-411: “To culture the Pf parasites, RBCs were isolated from whole blood, washed to remove residual plasma and exosomes, and cultured following the WHO-FIND-CDC Malaria Methods Manual 2018, SOP 4.1 Preparation of Reagents and Media for Culture of Malaria Parasites [43].

Major comment:

  • The authors state both in the Methods and the Results section that they performed a multiple comparison test. I don't understand how this was possible since there are 6 groups with 4 subjects each, leading to not enough degrees of freedom to perform multiple comparisons. Could they be clearer regarding the groups they compared? Did they perform multiple comparisons only in groups that provided statistically significant results from ANOVA?

We did post hoc analysis between groups because our F-test indicated differences between the means for our 6 groups. For this, we added the following sentence in line 91-93 as well as reference 25: “We completed post hoc multiple comparison tests to determine group differences while controlling for Type I error inflation [25].”

Results:

Minor comments:

  • To facilitate the reader, I believe that the authors should break Supplementary Figure 1 into two Figures. Since Supplementary Figure 1 panels E-P are mentioned after Supplementary Figure 2, I would suggest that the authors move these panels to a novel Supplementary Figure (namely, Supplementary Figure 3; and change the current Supplementary Figure 3 to Supplementary Figure 4) so that the reader doesn't have to go back and forth when reading the article.

Done.

2) In line 137 the authors state "on day 16, mainly gametocytes were observed" and afterward, line 141 starts with "the phases of gametocytes were also determined". I think the authors should first mention gametocytes when their results are shown so that the reader can follow the manuscript.

The following sentence was added in Lines 135-138: “Furthermore, the phases of gametocytes were observed each day (Figure 3). According to previous research [26], gametocytes can be classified into five distinct morphological phases (phases 1-5) in RBCs.”

3) I think that Table 1 should be transferred near the section it is first mentioned.

Moved to lines 171

Major comments:

4) Figure 2 is really hard to follow. Since I understand why it is structured in this way, I think the authors could add a line in the middle to separate the Results regarding the one miRNA from the other and maybe add a title above each side (e.g., the names of the miRNAs) to help the reader follow the Figure.

Figure 2 was changed to figure 4 (lines 231)

  • In all Figures legends the authors should remove the sentences "there was (not) a significant difference/correlation" and rather add a sign of significance on the diagrams, since the legends are extremely extensive and really hard to follow.

All figure legends have been extensively modified according to these suggestions.

  • A general comment: The authors spend a lot of time writing about Results that are not statistically significant. Please refrain from comparing values with no significance. For example, in lines 131-132 it is written "Still on day 8, there were no significant differences in schizont counts between the sickle Hb genotypes, but HbSS had the most and HbAS the least". Since there is no statistical significance, I don't find it proper to mention which genotype has the most and which the least. Or, at least, the authors are kindly asked to refrain from doing in excessively in the manuscript.

Changes were made through the entire manuscript.

Discussion (major comment):

Many parts of the Discussion section are just a brief repetition of the Results. I understand that in a manuscript with so many comparisons, it is useful to repeat the main findings in the Discussion, but I believe that the authors need to add more pieces that interpret their findings. In my opinion, they should try to explain their findings based on the current bibliography, and not just state that they are in line with previously reported data. For example, it is known that there are hematological and biochemical differences between these genotypes (10.1371/journal.pone.0228399), as well as a distinct oxygen delivery index (10.3233/CH-2008-1142). Indeed, oxygen has been found to play a role in malaria resistance (10.1073/pnas.1804388115). Moreover, in a similar background, namely beta-thalassemia trait, the different genotypes result in different redox equilibrium and distinct cytoskeleton structure depending on the severity of the mutation (10.3389/fphys.2022.907444). Could be this also true for the different sickle cell genotypes? Especially since both oxidative stress and cytoskeleton features have been linked to malaria resistance (10.1038/ncomms13401; 10.1002/(SICI)1096-8644(199906)109:2<269::AID-AJPA11>3.0.CO;2-#)? I believe that the authors could add some bibliographically-based hypotheses to explain their Results, like the suggestions above. Of course, they are free to search and reference more relative articles.

The suggested paragraph has been added to the discussion. We thank the reviewer for adding valuable and constructive insight to the text's relevance. Lines 361-369: ”The literature suggests that there are significant hematological and biochemical differences between sickle cell genotypes, which may result in varying oxygen delivery in-dices [34,35]. This is supported by previous research indicating that oxygen plays a role in malaria resistance [36]. Moreover, studies examining the impact of beta-thalassemia trait on redox equilibrium and cytoskeleton structure, depending on the severity of the mutation, suggest that similar differences may exist between sickle cell genotypes [37]. Given that oxidative stress and cytoskeleton features have been linked to malaria re-sistance, it is possible that these genotype-related differences may also play a role in sickle cell's protective effects against malaria [38,39].

Overall: There are many grammatical/syntax errors in the manuscript. The authors are kindly requested to proofread their Revised manuscript before its resubmission.

Most of the text was extensively changed for better English and to avoid the plagiarism for the thesis work.

Round 2

Reviewer 2 Report

The authors have addressed the concernss raised by me

Reviewer 3 Report

The authors have extensively altered their manuscript and all my comments have been exceptionally addressed. I believe that their manuscript can be accepted in its current form.